# Environmental Status and Human Health: Evidence from China

**DOI:** 10.3390/ijerph191912623

**Published:** 2022-10-02

**Authors:** Suli Cheng, Zubing Xiang, Haojun Xi

**Affiliations:** 1School of Mathematics and Statistics, Chongqing Technology and Business University, Chongqing 400067, China; 2School of Physical Education, Chongqing University, Chongqing 400044, China

**Keywords:** environmental status, mortality, deviation decomposition, semi-parametric additive panel model

## Abstract

In recent years, there have been growing concerns about the environment and its effect on human health. In this paper, we measure human health by mortality. Firstly, we use the method of deviation decomposition to investigate the different changes of mortality in eastern, central and western regions of China. Secondly, we study the linearity and nonlinearity between environmental status and mortality by semi–parametric additive panel model. Following is the primary conclusions obtained in the study: (1) There exists a big mortality gap among different regions; the gap is mainly dominated by the inter–regional difference; the mortality of the middle region increases heavily; the western region becomes a major source of mortality differences. (2) Mortality decreased with the increase of urban green area. On the other hand, the higher the environmental pollution index, the higher the mortality rate. (3) The environmental pollution index, urban green area, number of licensed (assistant) physicians per thousand and the per capita GDP can affect mortality in a nonlinear way.

## 1. Introduction

With the rapid development of the social economy and continual improvement of life quality, the environmental health becomes more and more concerned by countries all over the world. Serious ecological and environmental problems have brought the great danger to human health. According to the estimation of the World Health Organization (WHO), more than 24% of all deaths worldwide were attributable to the environment in 2016. In 2020, 208 environmental emergencies had occurred in China, down 2.5% year-over-year. Mortality is an important barometer to measure the human health. The National Bureau of Statistics of China shows that the mortality maintains 6‰–7‰ after China started its reform and opening-up. However, the mortality exceeds 7‰ from the beginning of 2008. At present, China has gotten the prominent period of environmental pollution health and frequent period of environmental health events. Therefore, it is great significance that we should study the influence of environmental factors on the mortality of China.

The study on the health crisis caused by environmental pollution is an attractive issue which caught scholars’ attention from all over the world. The related research mainly focuses on the fields of medicine, environment and economy. Research on the environmental effects on health from an economic point of view begins with health production functions in Grossman (1972) [1]. In these studies, an important aspect concerns the impact of environmental factors on mortality. In the next, we review the literature from micro-data and macro-data analysis respectively:

Dockery et al. (1993) [2] estimated the effects of air pollution on mortality. Survival analysis was conducted with data from a 14-to-16-year mortality follow-up of 8111 adults in six U.S. cities. They observed statistically significant and robust associations between air pollution and mortality. Mortality for the most polluted of the cities is 1.26 times as compared with the least polluted. Xu et al. (1996) [3] discussed the effects of air pollution in Shenyang city in China on mortality, acute and chronic disease used ecological and time series. They found that the total mortality, chronic obstructive pulmonary diseases and cardiovascular disease have significant impact on air pollution. Li et al. (2010) [4] reported findings on air quality and outpatient visits for asthma among adults in Beijing during the 2008 Summer Olympic Games. The results showed that even in a heavily-polluted city, decreased concentrations of small particles were associated with some reduction in asthma visits in adults. Tessum et al. (2021) [5] showed that nearly all major emission categories contribute to the systemic PM ${}_{2.5}$ exposure disparity experienced by people of color.

Franz and FitzRoy (2006) [6] studied child mortality and fertility in 61 developing countries including the Central Asian Republics (CARs) by the method of instrumental variables, then they found that the relative high rate of mortality in CARs was largely related to the environmental degradation in the region. Qi (2008) [7] analysed the relationship between the environment and population health on the basis of a simultaneous equations model using province-level panel data covering the 1990–2006 in China. The results indicated that emissions of various industrial pollutants had divergent trends and suggested that pollution impedes economic growth. In the last few decades, researchers have studied the relationship between environment and human health by using linear models. The current literatures explain various applications of the generalized additive model to link air pollution, climatic variability with adverse health outcomes. Ravindra et al. (2019) [8] highlight the relationship with ambient air pollutants, climate change and health by evaluating studies related to the generalized additive model.

Greenery as one of environment influence factors, it plays an important role on human health. Chang and Li (2014) [9] obtained that trees in general, and broadleaved trees in particular, play a major positive role in meliorating the urban environment. Even small groups of trees, or lines of them along a street, can reduce both noise and levels of pollution in the near-ground layers of air (Streiling and Matzarakis (2003) [10]). Dillen et al. (2012) [11] shown positive relationships between the objectively determined quantity of green areas in the neighbourhood and people’s self-rated health. Nevertheless, greenery also have a negative impact on health. Kuchcik et al. (2016) [12] pointed that plants in a warmer climate can produce larger amounts of pollen and are more readily able to initiate an allergic reaction among those prone to them. That means-not every greenery is advantageous for humans in the cities. Mitchell and Popham (2007) [13] suggested that future research on greenspace and health should include its quality, poor quality greenspace might even be detrimental to health. Therefore, the quantity and also the quality of greenspace in human health.

To sum up, air pollution, greenery area, social economic development level and so on have importance on human health. To the best of our knowledge, the research on the effects of human health using regional macro data and contained both the environmental greening and environmental pollution is relatively few. Therefore, the goal of this study is to analyze the linear and nonlinear effects of environmental factors on human health using semi-parametric additive panel model.

## 2. The Difference of Human Health in China

Xie et al. (2016) [14] evaluated the spatial distribution of premature deaths in China between 2000 and 2010 attributable to ambient PM2.5 based on a high resolution population density map of China, satellite retrieved PM2.5 concentrations, and provincial health data. In order to investigate the different changes of health status in eastern, central and western regions( The eastern includes 11 provinces and cities, which are Beijing, Tianjin, Hebei, Liaoning, Shangdong, Shanghai, Jiangsu, Zhejiang, Fujian, Guangong and Hainan; The central includes 8 provinces, which are Shanxi, Jilin, Heilongjiang, Anhui, Jiangxi, Henan, Hubei and Hunan; The western includes 12 provinces and cities, which are Neimenggu, Guangxi, Yunnan, Guizhou, Sichuan, Chongqing, Xizang, Shanxi, Gansu, Ningxia, Qinghai and Xinjiang. It did not include Taiwan, Hong Kong and Macau because of the difference in statistical criterion) of China, we used the mortality as an indicator to reflect overall human health. Moreover, we calculated the average mortality of the provinces in eastern, central and western regions and their differences between other provinces average mortality, according to the statistics data of mortality in 31 provinces, autonomous regions and municipalities from 2011 to 2019 of China. (The results are shown in Table 1). Table 1 shows that the mortality gap among each provinces in China remained around 5.84‰–6.21‰ since 2011; the eastern regional mortality level is all lower than the national average. However, the gap between the national average level was expanding. The central area mortality level was much higher than the other two regions in the most of the years. The western region is ranked in the middle of the three regions. There is no significant difference between the average level of the western region and the whole country.

After analyzing the differences in mortality between the level in each region and the national average level, we further decompose the sum of the squares deviations of mortality levels and analyze the internal differences in mortality across different regions and their contribution to the total variance. Suppose $X_{ij}$ is the mortality of the *j*th province (autonomous region or municipality) in the *i*th region, $\bar{X_{i}}$ is average mortality in the *i*th region, $\bar{X}$ is national average mortality, so the total sum of squares deviations $\sum_{i=1}^{3}\sum_{j=1}^{n_{i}}{\left(X_{ij}-\bar{X}\right)}^{2}$ could be decomposed into sum of squares deviations within classes $\sum_{i=1}^{3}\sum_{j=1}^{n_{i}}{\left(X_{ij}-\bar{X_{i}}\right)}^{2}$ and sum of squares deviations between classes $\sum_{i=1}^{3}n_{i}{\left(\bar{X_{i}}-\bar{X}\right)}^{2}$ in each region, specific as follows:(1)$$\begin{array}{c} \sum_{i=1}^{3}\sum_{j=1}^{n_{i}}{\left(X_{ij}-\bar{X}\right)}^{2}=\sum_{i=1}^{3}\sum_{j=1}^{n_{i}}{\left(X_{ij}-\bar{X_{i}}\right)}^{2}+\sum_{i=1}^{3}n_{i}{\left(\bar{X_{i}}-\bar{X}\right)}^{2} \end{array}$$
where $j=1,\ldots ,n_{i},i=1,2,3,n_{1}=11,n_{2}=8,n_{3}=12.$

We use the ratio of the sum of squares deviations within classes to the total sum of squares deviation of the whole country to represent the contribution degree of differences between the regional to total differences, and use the ratio of the sum of squares deviations between classes and the total sum of squares deviation of the whole country to represent the contribution degree of differences between the regional to total differences. Specific calculation results are shown in Table 2:

Table 2 show that there were the least differences and the highest-level homogenousity in mortality among the provinces in central China, while the greatest differences and the lowest-level homogenousity in the western region. The degree of differences among the provinces in the eastern region fell somewhere in between those of the other two areas, and got closer to that in central China. The relatively large mortality differences among the provinces in western China is possibly related to the vast territory, complicated geographical and climatic conditions, and the relatively backward economic development in this region.

Based on the analysis the average level of regional mortality and its contribution to the overall national differences, we found that the average level of mortality in the eastern region is lower than the national average, and the death status shows signs of some improvement compared to the overall situation of the whole country, and the internal differences are reduced. It is not optimistic that mortality in eight provinces of the central region in recent years, and internal differences are shrinking along with the gradual rising on mortality. The gap of western regions mortality and the national average has decreased. Comparing the overall situation of the whole country, mortality status has improved, but the improvement is not synchronous. Regional differences are gradually widening and the western region gradually becomes the main source of inter-regional difference.

## 3. Research Scheme Design

### 3.1. Variable Selection and Data Preprocessing

In this paper, we use the relevant indicators data of 31 provinces, municipalities and autonomous regions in China to make econometric modeling analysis. All the data are obtained from the Statistical Yearbook of China and China City Statistical Yearbook (2011–2020). Obviously, the death rate (‰, denoted as *mortality*) can be directly used as explained variable. Mortality is influenced by many factors, such as smoking (Janssen and Spriensma (2012) [15]), disease (Dolejs (2014) [16]), climate, air quality (Dockery et al. (1993) [2]), age, gender, region (Behl (2013) [17]), socio-economic (Kan et al. (2004) [18]) and so on. Based on the above discussion, the specific meaning of the explanatory variable as follows:

In aspects of core explanatory variables, that is, the environmental situation, the existing literature concerned more on the environmental pollution. Considering environmental greening is an inseparable part of environmental system, and the higher green coverage can improve the living environment, benefit to the physical and mental health (Dahlkvist et al. (2016) [19], Dillen et al. (2012) [11]). Therefore this paper will consider from two aspects of greening and pollution to the environmental factors. Regarding to greening situation, we use the urban green area (hectare/person, denoted as *greenland*. It refers to landscaping areas and various green areas consist of lawns and trees, the main area includes public green space, residential green space, unit green space, road green space and park green space) as an explanatory variable (Kan et al. (2004) [18]). As to the pollution, we use the information entropy(It is to assign a weight to the indicator, which can judge the dispersion degree of each indicator. The greater the dispersion degree, the smaller the entropy value, so that the data contain more information.) calculation method, and select three classes (waste water, waste gas and industrial solid wastes) environmental pollution indexes, the total volume of waste water discharge by each region (million tons, denoted as $\mu_{1}$, it includes the volume of production waste water and domestic sewage), the total emission of sulfur dioxide (million tons, denoted as $\mu_{2}$, total volume of waste gas emission refers to waste gas emitted from burning of fuels and from the production process, includes sulfur dioxide, nitrogen oxides particulate matter, volatile organic compounds and so on. This article uses sulfur dioxide emission instead of waste gas emission) and volume of industrial solid wastes produced (million tons, denoted as $\mu_{3}$, it refers to the total volume of solid, semi solid or high concentration liquid residues produced by industrial enterprises in their production process), to construct a comprehensive indicators (denoted as *polu*) to measure environmental pollution as a proxy variable (Peng and Bao (2006) [20]). The main reasons are: First, the environmental pollution is caused by the combined role effects of various pollutants, a separate consideration of a pollutant may be biased; Second, taking into account the frequent occurrence of smog and algal bloom phenomenon in recent years, causing serious air pollution and water pollution. Then, the pollution led to all kinds respiratory and cardiovascular diseases, which affected human health (Shafik (1992) [21]; Mead (2005) [22]); Third, to measure environmental pollution degree, some scholars use the pollutant emissions (Peng and Bao (2006) [20]), others use indication such as total suspended particles, $PM_{10}$ and $SO_{2}$ (Chen (2002) [23]; Kan (2004) [18]; Miao and Chen (2010) [24]). Something need to be explained, in 2011, the Ministry of Environmental Protection conducted a statistical institution revision, and its pollution indicators then composed of industrial sources, agricultural sources and town living sources. The specific construction method of the proxy variable is completed according to the following steps:

First step, carried on normalization processing of the data, we obtain
$$p_{ij}=\left(u_{ij}-\min_{i}\left\{u_{ij}\right\}\right)/\left(\max_{i}\left\{u_{ij}\right\}-\min_{i}\left\{u_{ij}\right\}\right),i=1,\ldots ,n,j=1,2,3,$$
where $\max_{i}\left\{u_{ij}\right\}$ and $\min_{i}\left\{u_{ij}\right\}$ indicate the maximum and minimum values in the *jth* initial evaluation indication respectively.

Second step, in order to avoid computing of impacting information entropy(it can usually be used as a measure of the complexity of a system. The more complex a system is, the more different kinds of situations occur, and then the larger information entropy is. On the contrary, the simpler a system is, the fewer different kinds of situations occur, and then the smaller information entropy is) in the case of indicator approach to zero, process the normalization indication, and plus a constant 1, we have $p_{ij}^{{}^{\prime }}=1+p_{ij}$.

Third step, calculating the numerical of information entropy in the *jth* initial evaluation indication
$$e_{j}=-\sum_{i=1}^{n}b_{ij}ln\left(b_{ij}\right)/ln\left(n\right),\;j=1,2,3,$$
where $b_{ij}=p_{ij}^{{}^{\prime }}/\sum_{i=1}^{n}p_{ij}^{{}^{\prime }}$.

Fourth step, calculating the weight of information entropy in the *jth* indication
$$w_{j}=\left(1-e_{j}\right)/\sum_{j=1}^{3}\left(1-e_{j}\right),\;j=1,2,3.$$

Fifth step, calculating the numerical of comprehensive pollution indication in the *ith* region
$$polu_{i}=\sum_{j=1}^{3}w_{j}p_{ij}^{{}^{\prime }},\;i=1,\ldots ,n.$$

When controlling variables, this paper mainly chooses the variables from the economic factors and health factors. In general, scholars often chose the per capita gross domestic product (denoted as *pgdp*) to replace GDP. Referring to the relevant researches (Kan et al. (2004) [18], Peng and Bao (2006) [20]), this paper selected 2011 as the base year, *pgdp* which obtained after indices treatment (million Yuan RMB/person) and number of licensed (assistant) physicians per thousand (people/thousand people, denoted as *doctor*) to measure of the economic development and proxy variables of medical conditions.

### 3.2. Semi-Parametric Additive Panel Model

Regression models are important tools in data analysis. The standard linear regression model assumes a simple form for this conditional expectation:$$E\left(Y|X_{1},X_{2},\ldots ,X_{p}\right)=\alpha_{0}+\alpha_{1}X_{1}+\ldots +\alpha_{p}X_{p}.$$

Given a sample, estimates of $\alpha_{0},\alpha_{1},\ldots ,\alpha_{p}$ are usually obtained by least squares, maximum likelihood and so on.

Environmental conditions and other factors’ influence on mortality may be complex. There is no doubt that it is a limitation on using linear parametric methods to characterize the impact on mortality. Addition model generalizes the linear regression model and model the dependence of *Y* on $X_{1},X_{2},\ldots ,X_{p}$ in a more nonparametric fashion. The cross-sectional nonparametric additive model and its estimation method were initially proposed by Stone (1985) [25].
$$E\left(Y|X_{1},X_{2},\ldots ,X_{p}\right)=\alpha +\sum_{k=1}^{p}g_{k}\left(X_{k}\right).$$

On this basis, scholars consider the following nonparametric additive panel model, which extends the cross-sectional nonparametric additive model. The model is defined by
(2)$$E\left(Y_{ij}|x_{ij1},\ldots ,x_{ijp}\right)=\alpha +\sum_{k=1}^{p}g_{k}\left(x_{ijk}\right),$$
where $Y_{ij}$ is the ij th observation of explained variable, $x_{ijk}$ was the observation of the *kth* explanatory variables at the *ith* individual in the *jth* moment. $g_{k}\left(\cdot \right)$ is an unknown function of one element, α is an unknown parameter, i=1,…,N,j=1,…,T,k=1,…,p.

Therefore, we intend to use the effect of linear and nonlinear between variables on the semi-parametric additive model simultaneously, to find more complicated nonlinear characteristics, which contribute to the further study of the relationship between the variables to ensure no loss of the linear information. In the formula (Equation 2), the linear part is added, and the semi-parametric additive model is obtained as follows:(3)$$E\left(Y_{ij}|x_{ij1},\ldots ,x_{ijp}\right)=\alpha +\sum_{k=1}^{p}\beta_{i}x_{ijk}+\sum_{k=1}^{p}g_{k}\left(x_{ijk}\right)$$

For the estimation of model (Equation 3), this paper uses spline function estimation, and its specific calculation methods and convergence are proved in Du et al. (2018) [26] and Wang and Yang (2007) [27]. Generally speaking, panel data models consider that the fixed effects model, random effects model and without any spatial interaction effects. A Hausman specification test may be used to solve the problem of model selection. The results are listed in the Table 3. Observed the Table 3, *p*-value is too small to choose the random effects model.

Now, we will analyze the influence factors of mortality in China using the semi-parametric additive panel model with fixed effect as follows:(4)$$\begin{array}{c} E\left(mortality|greenland,polu,pgdp,doctor\right)=\alpha_{0}+\beta_{1}greenland+\beta_{2}polu\\ +\beta_{3}pgdp+\beta_{4}doctor+g_{1}\left(greenland\right)+g_{2}\left(polu\right)+g_{3}\left(pgdp\right)+g_{4}\left(doctor\right), \end{array}$$
where conditional mean of mortality is on the left of the equation, $\beta_{i}$ is linear partially estimated parameter, $g_{i}\left(\cdot \right)$ is nonlinear part. Logarithmic transformation of variables *greenland* and *pgdp* to relieve the trouble caused by big gaps in the domain. The estimation results of the linear part of the model are presented in Table 4, and the nonlinear results are shown in Table 5.

From Table 4, we can get the explanatory variables are significant on the linear components except *doctor*. Similarly, we obtain the all explanatory variables are significant on the nonlinear components from Table 5. Therefore, the semi-parametric additive model with fixed effect can be represented in the form
(5)$$\begin{array}{c} E\left(mortality|greenland,polu,pgdp,doctor\right)=\alpha_{0}+\beta_{1}greenland+\beta_{2}polu\\ +\beta_{3}pgdp+g_{1}\left(greenland\right)+g_{2}\left(polu\right)+g_{3}\left(pgdp\right)+g_{4}\left(doctor\right) \end{array}$$

The analytical results of unknown linear parametric are listed in the Table 6. The function estimates of nonparametric components are displayed in the Table 7 and Figure 1. From the Table 6, we can get the conclusions as follows. Firstly, the linear effect is significant both for environmental pollution index (*polu*), urban green area (*greenland*) and per capita GDP (*pgdp*) on the mortality. Secondly, the impact of *greenland* is negatively significant and elastic coefficient is -0.5685, which means the influence of *greenland* is positive. It indicates that urban green areas (*greenland*) increasing will help to reduce the mortality. These results are consistent with the analytical conclusions of Dillen (2012) [11]. Thirdly, the coefficient of *polu* and *pgdp* is 0.4308 and 1.1096 respectively, which means a worsening of environmental pollution and per capita GDP will lead to a significant increase in mortality. Therefore, government and factories should effort to have strict control and governance over pollution sources.

The Table 7 and Figure 1 show that the nonlinear effect is significant all explanatory variables on the mortality. Each effective degree of freedom (edf) of spline function is also listed in Table 7. Figure 1 shows that the fitted results and 95% confidence intervals of unknown additive function, which reflects the existing complex nonlinear inner action of each factor on mortality.

Figure 1 has clearly shown that environmental pollution index (polu) has non-linear effects on mortality and such effect fluctuates smoothly; the influence of urban green area (greenland) on mortality presents the obvious inverted “U” non-linear movement; the effects of doctor (number of licensed physicians per thousand) on mortality displays the tendency of continuous decrease; the effect of per capita GDP (pgdp) on mortality swings in a non-linear way and move downwards on the whole. It can be known that the effective methods to reduce mortality is increasing greenland, doctor and pgdp, with the growth in doctor as one of the most direct, efficient and stable means. Besides, the rise in greenland cannot exert immediate effects on mortality but as a lagging indicator, so policy makers need to have a long-term strategic perspective so as to reduce mortality by pulling up greenland. The surge in pgdp can slash mortality overall, but its function represents non-linear swings and inclination of being vulnerable to other social factors, and particularly higher economic levels exert more remarkable impact on mortality. Therefore, with the enhancement of economic levels, governments of all countries should strengthen the regulations on health management of their people.

## 4. Conclusions

Firstly, this study provides details on the current situation and regional differences of mortality in China by means of descriptive statistics and deviation decomposition. Secondly, we make use of the panel data of 30 provinces, municipalities and autonomous regions in China from 2011-2019 and construct a semi-parametric additive panel model to examine how environmental conditions exert effects on mortality in both linear and nonlinear forms. The results are as follows: (1) mortality in China greatly varies from region to region, with intra-regional differences highlighted, a remarkable surge in mortality in the central part of China, and the change in mortality in western China as the main source of the overall differences. (2) polu, greenland, and pgdp have significant linear effects on mortality, and meanwhile polu, greenland, doctor and pgdp can affect mortality in a nonlinear way.

Based on the analysis above, we give the following recommendations in policies: (1) Step up efforts to have strict control and governance over pollution sources. The empirical results have proved that polu has relatively direct effects on mortality, so in the context of a soaring mortality in China, the government needs to redouble efforts to reduce pollution emissions and take various measures to cut the pollution grade as soon as possible after pollution occurs. (2) Attach greater importance to urban and rural landscaping. Although the level of the environment afforestation has limited effects on the relatively high mortality, such effects are far-reaching, with the result that the government needs to formulate the reasonable land policy of landscape construction, launch tailored greening programs, promote new technologies, and strengthen protective measures for forests and wetlands according to the ecological situations in different places. (3) Increase investments in medical and health resources and improve the medical and health service system. The research on how medical conditions affect mortality demonstrates that the current expansion of medical resources in China has a significantly inhibiting effect on mortality, so the improvement in medical and health services can make remarkable achievements in the elimination of health risks imposed by environmental pollution on the Chinese residents.

## Figures and Tables

**Figure 1 ijerph-19-12623-f001:**
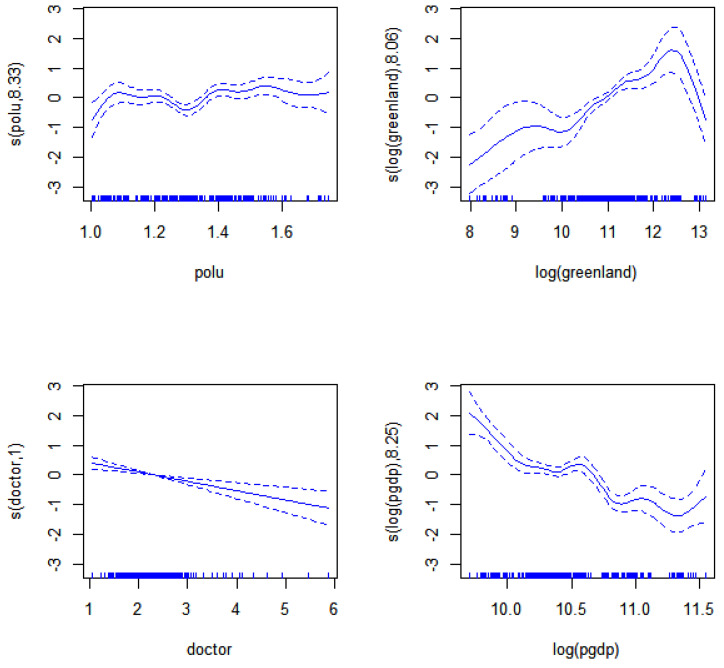
The nonlinear effect of environmental status on mortality.

**Table 1 ijerph-19-12623-t001:** Mortality levels and differences of nation and each region (unit: ‰).

	Nation		Eastern Region		Central Region		Western Region	
	**Mean**	**Std**	**Mean**	**Differences in the Nation**	**Mean**	**Differences in the Nation**	**Mean**	**Differences in the Nation**
2011	5.84	0.76	5.65	−0.20	6.05	0.20	5.89	0.05
2012	5.97	0.79	5.85	−0.12	6.17	0.20	5.95	−0.02
2013	5.98	0.74	5.84	−0.15	6.11	0.13	6.03	0.05
2014	6.11	0.73	5.90	−0.21	6.45	0.34	6.08	−0.03
2015	5.98	0.80	5.78	−0.19	6.20	0.22	6.01	0.03
2016	6.07	0.83	5.92	−0.15	6.37	0.30	6.00	−0.06
2017	6.14	0.84	5.99	−0.15	6.45	0.31	6.08	−0.07
2018	6.17	0.81	6.07	−0.11	6.39	0.22	6.12	−0.05
2019	6.21	0.86	6.04	−0.17	6.60	0.39	6.11	−0.10

**Table 2 ijerph-19-12623-t002:** Contribution degree of each regional mortality differences on national total differences (unit: %).

	Contribution Degree of Differences within the Region				Contribution Degree of Differences between the Region
	Eastern Region	Central Region	Western Region	Total	
2011	44.30	8.25	42.87	95.42	4.58
2012	40.69	9.58	47.19	97.46	2.54
2013	39.21	15.88	42.5	97.59	2.41
2014	34.07	8.99	47.89	90.95	9.05
2015	33.94	12.27	49.58	95.78	4.22
2016	36.02	13.98	45.09	95.09	4.91
2017	40.02	11.49	43.47	94.98	5.02
2018	39.03	12.97	45.21	97.20	2.8
2019	38.16	9.23	45.25	92.65	7.35

**Table 3 ijerph-19-12623-t003:** The results of Hausman test.

$\chi^{2}$	df	*p*-Value
82.78	4	<2.2 ×$10^{-16}$

**Table 4 ijerph-19-12623-t004:** The linear effect result of environmental status on mortality.

Variable	Estimation	Std	t-Value	*p*-Value
Intercept	0.1014 ***	0.0260	3.900	<0.0001
polu	0.4293 **	0.1312	3.271	0.0012
Log(greenland)	−0.5756 **	0.1938	−2.971	0.0033
doctor	0.0838	0.0540	1.553	0.1216
Log(pgdp)	1.0987 ***	0.1956	5.617	<0.0001

** *p* < 0.01, *** *p* < 0.001.

**Table 5 ijerph-19-12623-t005:** The nonlinear effect result of environmental status on mortality.

Variable	edf	Ref.df	t-Value	*p*-Value
S (polu)	8.3338 ***	8.8515	5.324	<0.0001
S (Log(greenland))	8.0546 ***	8.5740	10.549	<0.0001
S (doctor)	0.7325 ***	0.7325	55.998	<0.0001
S (Log(pgdp))	8.2432 ***	8.7898	11.239	<0.0001

*** *p* < 0.001.

**Table 6 ijerph-19-12623-t006:** The linear effect result of environmental status on mortality.

Variable	Estimation	Std	t-Value	*p*-Value
Intercept	0.1026 ***	0.0267	3.840	<0.0001
polu	0.4308 **	0.1314	3.280	0.0012
Log(greenland)	−0.5685 **	0.1929	−2.947	0.0035
Log(pgdp)	1.1096 ***	0.1965	5.646	<0.0001

** *p* < 0.01, *** *p* < 0.001.

**Table 7 ijerph-19-12623-t007:** The nonlinear effect result of environmental status on mortality.

Variable	edf	Ref.df	t-Value	*p*-Value
S (polu)	8.334 ***	8.851	5.319	<0.0001
S (Log(greenland))	8.059 ***	8.578	10.538	<0.0001
S (doctor)	1.000 ***	1.000	15.628	<0.0001
S (Log(pgdp))	8.245 ***	8.792	11.249	<0.0001

*** *p* < 0.001.

## Data Availability

All the data are obtained from the Statistical Yearbook of China and China City Statistical Yearbook (2011–2020). It is available on request from the corresponding author.

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
