# Peer review of "Environmental Status and Human Health: Evidence from China"

_ijerph, 2022, doi:10.3390/ijerph191912623_

Round 1

Reviewer 1 Report

The manuscript is clear, relevant to the field, and presented in a well-structured manner. The design of the experiment is appropriate to test the hypothesis presented in the paper. Tables and figures present the data correctly, are easy to  analyze. The conclusions are in line with the evidence and arguments presented. The current overview is still relevant and of interest to the scientific community. The results presented in the work show the progress of current knowledge, all conclusions are substantiated and supported by the results. The methods, tools, software and reagents are described in sufficient detail to allow another investigator to replicate the results. The cited literature is mainly new publications and relevant to the analyzed issue.

The work expands current knowledge, the conclusions are of interest to the readers of the journal.

In the second chapter, you only break down into geographical regions. Do the cultural or environmental elements of the given zones of the country affect the health level of the inhabitants? I am asking for one sentence of comment on this issue, as I believe that its omission is a significant omission. This is confirmed by using only the average, random element in formula (1), not averaged, for example, by weight to take into account additional factors. Each analysis presupposes cognitive criteria in nature. Therefore, the evaluation criteria adopted in Chapter 3 are acceptable. Additional could be included, but this would require additional analysis. Please comment why such evaluation criteria were adopted. This is related to the first question.

Stoun's estimation takes into account the weights mentioned in the formula 2 and 3, so please provide a brief description of the Stoun formula in relation to the analyzed case. These are the elements that are the starting point for your analysis in the article. So I am asking for a short description of them.

I agree with your general conclusions. However, quantitative analysis could take into account the impact of data acquisition errors. It is so important that the data come from the period from 2011 to 2019. There may have been elements influencing the data acquisition, analysis and interpretation. This is important in particular with the non-linearity of the course of changes in Fig. 1.

Author Response

The point-by-point response to the reviewer's comments is upload as a PDF file.

Reviewer 2 Report

The article raises a very interesting issue, but requires a short review of the impact of greenery on human health. The introduction in this regard is insufficient. Greenery can also have a negative impact on health (e.g. allergies ...).

The issue is very complicated and complex. Mortality is influenced by many factors, although, of course, analyzing all of them is an impossible task.

The article should emphasize the complexity of the issue, specify what else may have an impact on mortality, and explain the general approach used in more details.

The introduction lacks reports on the positive impact of greenery on human health, and such articles are available. Please supplement this part of the article with information based on the literature on the impact of greening on human health and the environment.

Description of environmental pollution indexes are very poor and required more details.

There is a need for some comments about used data about amount of wastewater, solid waste and greenings. How can you comment the raw data?

Please specify scientific definition of information entropy.

Line 9 – what certain size? In discussion I cannot find information about this size…

Line 24 – “hot” issue? It is scientific article, not popular science article… change for more suitable expression

Lines 26,27 and 48, 49 - please check the grammar

Page 2 – remarks – maybe it will be valuable to add a map?

Line 61 – punctuation, 2011 – 2019 31

Line 62 – showned?

Line 62 – it can be seen from table – change expression for more elegant

Line 68 – 70 sentence seems to be defective

Line 83 – 84 why? What do you think? Could you add some comments? What could be the reason? (year after year – maybe it is just a shift?)

Line 104 - please provide examples of this literature

Line 111 – please specify if this wastewater were treated or no, and what kind of treatment were used (only mechanical or also biological)? Why the unit is ton? What kind of wastewater – industrial or municipal ?

Line 112 – solid waste from which sector – please specify it,

Note that wastewater and solid waste (without industrial sector) are typically proportionally to the number of inhabitants

Line 126 - please make sure no parentheses are omitted, at least one…

Line 134 – what are the relevant researches? Please specify and add to the literature…

Author Response

(The authors gave the same response as above.)

Round 2

Reviewer 2 Report

I am simply repeating my review text as it has not been taken into account. Only the cosmetic corrections listed for individual lines have been included (anyway, without indicating new line numbers, which makes the subsequent correction significantly more difficult).

Such revision is simply a waste of time for editors and reviewers. Please take corrections seriously or try in another/this journal after major revision.

Moreover, the introduced amendments are sketchy and not very precise. They do not explain the indicated issues sufficiently.

For the sake of convenience, it would be preferable to add new lines, where there are changes to the article, and not leave the old numbering only.

The greenery  should be described – what are there – only lawns or tress? …..

“The article raises a very interesting issue, but requires a short review of the impact of greenery on human health. The introduction in this regard is insufficient. Greenery can also have a negative impact on health (e.g. allergies ...).

The issue is very complicated and complex. Mortality is influenced by many factors, although, of course, analyzing all of them is an impossible task.

The article should emphasize the complexity of the issue, specify what else may have an impact on mortality, and explain the general approach used in more details.

The introduction lacks reports on the positive impact of greenery on human health, and such articles are available. Please supplement this part of the article with information based on the literature on the impact of greening on human health and the environment.

Description of environmental pollution indexes are very poor and required more details.

There is a need for some comments about used data about amount of wastewater, solid waste and greenings. How can you comment the raw data?

Please specify scientific definition of information entropy.”

And some new comments:

Line 9 – what certain size? In discussion I cannot find information about this size…

Line 9 – in introduction ok, but in the text of article there is a need to comment and specify this “certain” size… what exactly it is?

Line 69 – 70, rather “…municipalities of China…”

Line 83 – 84 why? What do you think? Could you add some comments? What could be the reason? (year after year – maybe it is just a shift?)

However, this is a scientific article and it should be commented that it may only be a shift, not an unusual situation, and that it requires further observation, not a statement of dry facts. After all, this is probably not a report, but a scientific article ????

Line 111 – please specify if this wastewater were treated or no, and what kind of treatment were used (only mechanical or also biological)? Why the unit is ton? What kind of wastewater – industrial or municipal ?

Current line 120: please specify it in the article, there is a big difference between discharge of waste water and industrial solid waste….

But you should take also the detailed description from statistical yearbook about this value _ common industrial solid waste;

Line 112 – solid waste from which sector – please specify it – but in the article not only in the response to the reviewer…

Note that wastewater and solid waste (without industrial sector) are typically proportionally to the number of inhabitants… so maybe number of inhabitants should be also added…

Line 126 - please make sure no parentheses are omitted, at least one…

Now it is number 136 – no change has been done… still there is a lack at least one parenthesis…..

Author Response

(The authors gave the same response as above.)
